# Next-Generation Sequencing (NGS) Analysis Illustrates the Phenotypic Variability of Collagen Type IV Nephropathies

**DOI:** 10.3390/genes14030764

**Published:** 2023-03-21

**Authors:** Miriam Zacchia, Giovanna Capolongo, Francesca Del Vecchio Blanco, Floriana Secondulfo, Neha Gupta, Giancarlo Blasio, Rosa Maria Pollastro, Angela Cervesato, Giulio Piluso, Giuseppe Gigliotti, Annalaura Torella, Vincenzo Nigro, Alessandra F. Perna, Giovambattista Capasso, Francesco Trepiccione

**Affiliations:** 1Department of Medical and Translational Sciences, University of Campania, Luigi Vanvitelli, 81100 Caserta, Italy; 2Department of Precision Medicine, University of Campania, Luigi Vanvitelli, 81100 Caserta, Italy; 3Biogem, Scarl, 83031 Ariano Irpino, Italy; 4UOC Nefrologia e Dialisi, Ospedale Civile di Eboli “MM.SS. Addolorata”, 84025 Eboli, Italy; 5Telethon Institute of Genetics and Medicine, 80078 Pozzuoli, Italy

**Keywords:** basal membrane, *COL4*, kidney disease, NGS

## Abstract

Mutations in *COL4A3-A5* cause a spectrum of glomerular disorders, including thin basement membrane nephropathy (TBMN) and Alport syndrome (AS). The wide application of next-generation sequencing (NGS) in the last few years has revealed that mutations in these genes are not limited to these clinical entities. In this study, 176 individuals with a clinical diagnosis of inherited kidney disorders underwent an NGS-based analysis to address the underlying cause; those who changed or perfected the clinical diagnosis after molecular analysis were selected. In 5 out of 83 individuals reaching a molecular diagnosis, the genetic result was unexpected: three individuals showed mutations in collagen type IV genes. These patients showed the following clinical pictures: (1) familial focal segmental glomerulosclerosis; (2) end-stage renal disease (ESRD) diagnosed incidentally in a 49-year-old man, with diffuse cortical calcifications on renal imaging; and (3) dysmorphic and asymmetric kidneys with multiple cysts and signs of tubule–interstitial defects. Genetic analysis revealed rare heterozygote/compound heterozygote *COL4A4-A5* variants. Our study highlights the key role of NGS in the diagnosis of inherited renal disorders and shows the phenotype variability in patients carrying mutations in collagen type IV genes.

## 1. Introduction

The glomerular basement membrane (GBM) integrity is critical for glomerular filtration. Many unique structural components allow GMB to contribute significantly to the size selectivity of the glomerular filter. Among these components, type IV collagen is the most abundant. Moreover, collagen type IV chains are major components of basement membranes in the cochlea and eye. In the adult kidney, the GMB is mostly formed by α3, α4, and α5 intertwined subunits. The α5 subunit is encoded by *COL4A5*, located on the X-chromosome, and is associated with the classic X-linked Alport syndrome (AS) in male patients. The α3 or α4 subunits are encoded by *COL4A3* and *COL4A4*, respectively; their mutations cause either autosomal dominant/recessive Alport syndrome and thin basement membrane nephropathy (TBMN) [1,2]. Typically, AS patients’ phenotype is characterized by hematuria from infancy, with subsequent onset of glomerular filtration rate (GFR) decline up to end-stage renal disease (ESRD) in young-adults; extra-renal features include sensorineural hearing loss and sometimes lenticonus or retinal/corneal dystrophy [3]. Conversely, thin basement membrane nephropathy (TBMN) is characterized by a milder renal phenotype, with hematuria and sometimes proteinuria [4]. Affected individuals may have a uniformly thinned GMB, normal GFR, and a family history of hematuria. Their clinical course is usually benign. Sometimes, TBMN represents the carrier state for autosomal recessive AS. Genetic analysis to detect *COL4A3* and *COL4A4* mutations is not routinely performed in the diagnosis of TBMN, and technical difficulties may occur in variants’ interpretations. Recent advances in the genetic field have demonstrated that the spectrum of kidney dysfunction caused by collagen type IV mutations is not limited to the described clinical entities [5]. The present study shows the variability in the kidney phenotypes of patients carrying COL4 mutations, ranging from cystic disorders to glomerular barrier structural anomalies leading to proteinuria and progressive renal dysfunction. This study was conducted on 176 patients that underwent genetic analysis; in most patients, genetics confirmed the clinical suspicion, especially in patients with polycystic kidney disease or syndromic ciliopathies, as Bardet–Biedl syndrome. Interestingly, 5 out of 83 patients who had a molecular diagnosis showed unexpected results. Among these, three patients showed mutations in collagen type IV (COL4) genes. The study suggests that COL4 variants may underlie variable structural and functional kidney abnormalities, including both glomerular and tubulointerstitial disorders.

## 2. Methods

### 2.1. Clinical Information

Adult patients referred to our Center of Rare Kidney diseases that underwent genetic analysis in the Unit of Genetics of the University of Campania, Luigi Vanvitelli, were recruited.

All procedures performed in this study were in accordance with the ethical standards of the University of Campania “Luigi Vanvitelli” research committee (# 0017030/i-13/07/2020) and with the 1964 Helsinki declaration and its later amendments and informed consent was obtained from all individual participants included in the study.

We collected data from the genetic analysis of 176 patients suffering from genetic kidney disorders that showed a broad spectrum of phenotypes divided into the subgroups shown in Figure 1.

Clinical information was obtained from medical records of our Unit of Nephrology; in addition, two individuals were screened in the “Maria SS. Addolorata” hospital’s Unit of Nephrology (Eboli, SA, Italy). Experienced nephrologists performed physical examinations and assessments. Information including age, family history, and history of gestation was collected. Patients with a probable acquired pathogenesis (diabetes, probable immune disorders, and so on) were excluded from genetical analysis.

The suspicion of inherited kidney disorders was based on the criteria described below.

Autosomal dominant polycystic kidney disease (ADPKD) was diagnosed evaluating the number of kidney cysts at ultrasound and/or abdominal CT and familial history of kidney cysts, according to PEI criteria [6]. If no family history was present, patients were included only in the presence of imaging tests for enlarged multicystic kidneys suggestive of ADPKD.

Among cystic patients, 40 patients fulfilled the clinical criteria of Bardet–Biedl syndrome (according to Beales). Patients with glomerulonephritis underwent genetic analysis in case of familial forms and steroid resistant nephrotic syndrome. The latter was defined by the presence of oedema, massive proteinuria (>3.5 g/24 h) and hypoalbuminemia (<3 g/dL).

Alport syndrome (also included in the glomerulonephritis’ group in Figure 1), was diagnosed following current guidelines [7].

Concerning tubulopathies, hypokalemic tubulopathies were diagnosed in case of metabolic alkalosis and hypokalemia (after excluding a gastrointestinal or endocrine etiology), distal renal tubular acidosis (dRTA) was diagnosed with the urine acidification test, and Fanconi syndrome was diagnosed when metabolic acidosis, aminoaciduria and low molecular weight proteinuria were present.

Monogenic forms of urolithiasis were indagated in the case of family history, childhood onset, frequent recurrences, and nephrocalcinosis.

In total, 104 patients with cystic diseases (64 with polycystic kidney and 40 with Bardet–Biedl syndrome), 29 with glomerulonephritis, 5 with kidney stones and metabolic diseases, and 38 with tubulopathies were selected for genetic analysis. Their ages ranged from 18 to 62 years old. Ninety-four patients were female and 84 were male.

### 2.2. Molecular Analysis

Patients’ DNA was extracted from peripheral whole blood obtained using the QIAamp DNA Blood Kit by Qiagen. DNA quality and quantity were evaluated with spectrophotometry (Nanodrop ND 1000, Thermo Scientific Inc., Rockford, IL, USA) and fluorometry-based (Qubit 2.0 Fluorometer, Life Technologies, Carlsbad, CA, USA) methods. Patients were analyzed through NGS, 107 using a kidney-focused genetic panel (Nephroplex) containing 115 genes causing kidney diseases and the other 69 using Clinical Exome Sequencing, containing all disease-causing genes (approximately 5000). All analyzed genes have a clinical association with disease. Analysis was conducted using as a strategy the HaloPlex TM Target Enrichment System (Agilent)—using SureSelect Custom DNA Target Enrichment Probes, UNSPSC Code 41116134—followed by sequencing—using the HiSeq1000 system (Illumina inc., San Diego, CA, USA), as described elsewhere [8].

### 2.3. Variants Interpretation

After filtering variants for quality and number of reads (at least five reads for each variant), population databases (ExAC, gnomAD, and an internal database) [9,10] were used to filter variants according to population frequency and only rare alleles (minor allele frequency <1%) were included. In silico tools including SIFT, FATHMM, MutationAssessor, Polyphen-2, MutationTaster and Provean, MuPRO, PANTHER, PhD-SNP, and SNP n GO [11,12,13,14,15,16,17,18] were used to predict mutation pathogenicity; detected mutations were searched in the Clinvar database and familial segregation was performed when possible. Interpretation of variant pathogenicity followed the guidelines of The American College of Medical Genetics and Genomics (ACMG) [19]. The supposed causative mutations were confirmed by Sanger sequencing, as described elsewhere [20].

### 2.4. Patient Selection

Patients’ NGS reports were carefully analyzed by a multidisciplinary team including nephrologists and geneticists. All patients with a disease-causing mutation were analyzed to assess the correspondence between clinical diagnosis and molecular diagnosis. Patients with a discrepancy between clinical and molecular diagnosis were selected for a deeper analysis.

## 3. Results

### 3.1. Patient Cohort

A total of 176 subjects underwent genetic analysis with NGS technology using a kidney-focused genetic panel (Nephroplex) [2] (see additional data given in Online Resource 1) or exome sequencing. Disease-causing variants were found in 83 individuals (47.2%). As indicated in Figure 1, in 78 (94%) cases the genetic analysis confirmed the clinical diagnosis, whereas in 5 cases the genetics resulted in unexpected results. Three of these individuals showed a collagen type IV nephropathy. The composition of the cohort is briefly described in Table 1, and the diagnostic rate of the entire cohort is depicted in Figure 2. As shown, the patient cohort included kidney cystic individuals, syndromic patients, metabolic disorders, tubulopathies, and familial glomerulonephritis. The diagnostic rate was higher in patients with cystic and metabolic disorders (Figure 2).

### 3.2. Clinical Features of COL4 Patients

The clinical information and the genealogical tree of the five cases are shown in Table 2 and Figure 3, respectively.

Case 1 was a 43-year-old male patient with personal history of microhematuria during childhood that disappeared later, referring to the Nephrology Unit for the onset of proteinuria; a familiar history of chronic kidney disease was ascertained. His mother was diagnosed as having nephrotic syndrome due to focal and segmental glomerulosclerosis (FSGS). Moreover, three uncles were under dialysis treatment from a young age, suggesting the presence of an inherited nephropathy. At presentation, the patients showed an estimated glomerular filtration rate (eGFR) of 60 mL/min/1.73 m^2^. Proteinuria was 2 g/24 h. The screening for secondary forms of glomerulonephritis was negative. The patient underwent kidney biopsy, which revealed a classic picture of FSGS, with IgM and C3 mesangial deposits (Figure 4). Considering his personal and familiar story of kidney disease, the case and his mother underwent genetic analysis.

Case 2 was a 49-year-old man that referred to the hospital in emergency for epistaxis who had unremarkable personal and familial history of disease. He denied suffering from any significant morbidity; he was not under chronic medication and did not exhibit previous medical reports. On admission, high blood pressure and advanced renal dysfunction (estimated GFR of 13 mL/min/1.73 m^2^) were revealed. Sensorineural deafness was referred. Renal ultrasound showed a decreased size of the right kidney, with parenchymal calcifications and fetal lobulation bilaterally. The abdomen computed tomography (CT) scan confirmed cortical calcification (Figure 5). Subsequent analysis revealed that circulating parathyroid hormone (70 pg/mL), calcium (9 mg/dL, 2.25 mmol/L), and phosphate (4.1 mg/dL) levels were unremarkable considering the level of chronic renal disease. No vitamin D and/or calcium supplementation had been taken in the past. Few months later, the patient started chronic hemodialysis and, after two years, he underwent kidney transplantation. Because of the relatively young age at ESRD and the unknown cause of renal dysfunction, the patient underwent genetic analysis.

Case 3 was a 59-year-old male admitted to the Nephrology Unit for chronic kidney disease of unknown cause. The patient had suffered from polyuria and microhematuria since childhood; proteinuria appeared when he was thirty-one years old. In his familial history, the mother had intermittent hematuria, with normal overall renal function. His son showed hyperuricemia and a slight decline of eGFR. The proband underwent abdomen ultrasound and CT scan: the right kidney showed a reduced longitudinal diameter (10 cm) and subcentimetric cysts and four large cysts were detected on the left kidney (Figure 5B). Increased echogenicity with reduced corticomedullary differentiation were evident on ultrasound. The representation of the vascular tree was reduced. Because of the familiar history of renal disorders, the genetic analysis of the patient and his son was performed.

### 3.3. Genetics Analysis of Patients

Genetic variants found in patients are shown in Figure 6 and in Table 3.

For missense mutations we also calculated and reported the effect on protein structure according to SIFT. SIFT is a software that predicts the effect of aminoacidic changes on the protein structure using sequence homology and aminoacidic properties. For this reason, it is only applicable to non-synonymous variants. MuPro is a vector machine-based tool that predicts changes in protein stability for a single amino acid mutation. Panther measures PSAP (position-specific evolutionary preservation) to calculate the probability of the deleterious effect for missense mutations. PhD-SNP generates a comparative conservation score of multiple sequence alignment to identify the effect of the SNP as disease-related or neutral. SNP n GO is a support vector machine (SVM)-based method that uses information from multiple sequence alignment and gene ontology to predict if a given mutation is classified as disease-related or not. The effect of splicing of the first variant was predicted through SpliceAI [21].

Case 1 was studied with his mother. Both carried the rare heterozygote c.693+2T>C variant in COL4A4(NM_000092). The phenotype segregated with the genotype, as both had a history of nephrotic syndrome along with the histological diagnosis of FSGS. The mutation is predicted to affect protein splicing and is considered damaging by in silico programs. After genetic analysis, the proband and his mother were screened for ocular and auditory function; both were negative.

Molecular screening of Case 2 revealed the presence of the heterozygote c.991G>A COL4A5 variant, resulting in a p.Gly331Ser change. The variant is predicted as pathogenic by in silico programs such as SIFT, MuPRO, PANTHER, PhD-SNP, and SNP n GO.

Case 3 showed the c.1589G>A (p.Gly530Glu) mutation in COL4A4, predicted as likely to be pathogenic; the same mutation was also reported in his son, showing a similar renal phenotype.

## 4. Discussion

This paper shows unexpected clinical presentations of patients carrying *COL4A4-A5* mutations, suggesting that the spectrum of renal phenotypes associated with these genetic loci is wider than has been believed until now [7,22].

Collagens constitute a superfamily of structural proteins; 28 different subtypes have been described to date in vertebrates [23]. All of them, including collagen type IV, have a triple-helical structure composed of three α-chains. There are six different types of α-chains with similar structural domains. These include a short non-collagenous NH2-terminal domain (known as 7S domain), a long central collagenous domain composed of repeats of Gly-X-Y aminoacidic triplets, and a COOH-terminal domain (NC1 terminus).

Collagen type IV chains participate in the composition of basal membranes, whose defects are known to cause kidney dysfunction. In the human kidney, the α1 and α2 chains of collagen type IV are detected during embryonic development and their expression gradually decreases over time. Conversely, the α3(IV), α4(IV), and α5(IV) chains form the adult kidney basal membranes [24].

Mutations of type IV collagen are associated with the following human disorders: (1) AS, a progressive renal disorder characterized by structural abnormalities of the glomerular basement membrane (GBM) leading to hematuria and eGFR decline that is often associated with hearing loss and ocular symptoms and (2) TBMN, characterized by thinned GBM leading to hematuria and (sometimes) proteinuria. Moreover, an autoimmune disorder due to anti-α3(IV) antibodies (binding to alveolar and renal GBM) causes Goodpasture syndrome [4,25].

Recent studies have suggested that mutations in COL4A3-5 may also cause additional renal abnormalities [22]. However, little information is available on the relationship between cortical calcifications and *COL4* variants and very few data are available on the reasons for tubulointerstitial defects, including cysts.

Alport syndrome is the best-known nephropathy caused by collagen type IV mutations. It is caused by mutations in *COL4A5*, leading to X-linked Alport syndrome (XLAS) for 85% of cases [22,26]. Whereas the phenotype of female individuals is generally mild, male XLAS patients show progressive renal failure, especially when carrying truncating mutations. Extra-renal manifestations include hearing loss and ocular lesions, uncommon findings in hemizygote female carriers [27]. Biallelic mutations in *COL4A3* and *COL4A4* genes cause the autosomal recessive forms of AS, accounting for 15% of all AS patients, with a phenotype resembling male XLAS patients. Heterozygote mutations in *COL4A3* and *COL4A4* have been reported in autosomal-dominant AS and in TBMN.

Recently, NGS studies have demonstrated that mutations in these three genes may also result in atypical clinical presentations. Malone et al. performed whole-exome sequencing of a cohort of 70 families with a diagnosis of familial FSGS of unknown cause. In their study, seven families (10%) showed rare or novel *COL4A3* and *COL4A4* variants [28]. Additional studies have shown that familial FSGS is associated with mutations in COL4 genes, suggesting that GBM abnormalities due to defects in collagen IV chains may result in secondary FSGS [29]. Moreover, some reports have shown that *COL4A3* and *COL4A4* mutations have been detected in patients with multiple renal cysts [30,31,32].

The present study shows a series of patients carrying pathogenic variants in *COL4A4-A5* that show intriguing clinical presentations, expanding our understanding of the role of abnormalities in collagen type IV into the pathogenesis/predisposition to develop kidney disease. In this study, NGS analysis has been applied to 176 individuals to elucidate the molecular landscape underlying several classes of inherited renal disorders, including cystic, tubular, and glomerular disorders.

We selected patients whose diagnosis was unexpected. A high prevalence of these patients showed *COL4A4-5* mutations. The patients showed the following clinical pictures: familial FSGS (Case 1), cortical calcification with ESRD of unknown origin (Case 2), and familial dysmorphic kidneys with multiple cysts and renal function decline in adulthood (Case 3).

Case 1 underwent an analysis with his mother. Both had a biopsy-proven diagnosis of FSGS. They both showed the c.693+2T>C (NM_000092) COL4A4 variant. The detected mutation is predicted to affect protein splicing and is considered damaging according to in silico programs. The latter has never been reported in the literature nor in the public database ClinVar [33]. Even though no confirmation of functional abnormalities has been ever provided, the position of the mutation suggests that it could interfere with normal splicing, affecting a donor splice site.

Multiple in silico tools predicted that Case 2 and Case 3 harbor pathogenic missense variants as pathogenic; similar tools have also been utilized to predict pathogenic variants of other genes related to kidney disorders [34]. Case 2 showed the hemizygous missense c.991G>A p.Gly331Ser COL4A5 (NM_000495) variant. Public databases do not report this variant. It has been largely demonstrated that the conserved Gly residues are crucial components of collagen’s primary sequence. Given their small size and conformational flexibility, they are thought to play a role in the stability of the triple-helical structure [35]. Accordingly, glycine mutations are considered hot spots in the collagenous genes. Plant KE et al. reported the c.992G>T p.Gly331Val mutation classified as pathogenic. Moreover, in 2014 Hashimura Y et al. reported the c.991-1G>A mutation, which is considered pathogenic in public databases. As these mutations are located in the same region of Case 2, we assumed that it could affect protein function and the patient’s phenotype [36,37]. Thus, it is not excluded that the mutation in our patient perturbs normal splicing. Functional evidence regarding mRNA is needed to confirm this hypothesis. The patient’s renal phenotype is of high interest. Few reports in the literature have shown whether *COL4* mutations cause cortical calcification. The term nephrocalcinosis has been introduced to define generalized calcium deposition in renal parenchyma [38,39]. Cortical calcification is about 20 times less common than medullary calcification. According to the literature, it is the consequence of acute cortical necrosis [38]; however, chronic glomerulonephritis may sometimes cause this uncommon finding on imaging. On ultrasound, the patient showed increased echogenicity of the renal cortex. The CT scan confirmed this finding: atrophy of right kidney and bilateral thin circumferential cortical calcification.

The last patient (Case 3) showed multiple kidney cysts. Focal renal cysts are common in older subjects. The prevalence, size, and number increase with age. The exact pathogenesis is unknown. However, it is not excluded that weakening of the basement membrane along the nephron for different causes may induce or favor cysts formation [40]. The patient under examination showed centimetric cysts in the right kidney and large cysts on the left.

The analysis of *PKD1* and *PKD2*, the causative genes of autosomal dominant polycystic kidney disease [41], failed to show any pathogenic variant. This was not surprising as the imaging was atypical. Even if the number of bilateral cysts was quite high, kidneys were asymmetric, with a hypomorphic kidney with subcentimetric cysts and a contralateral enlarged kidney with multiple large cysts. This patient and his son showed the hemizygous c.1589G>A p.Gly530Glu variant in *COL4A4*, not reported in ClinVar but affecting glycine and therefore potentially impairing protein structure. In patients with *COL4A3-5*, the pathogenesis of tubulointerstitial nephritis and cyst formation is unclear [32,42]; some studies support the hypothesis that proteinuria may lead to progressive tubulointerstitial fibrosis. Whereas most investigations are focused on the alterations of the GBM, little is known about the pathogenesis and changes in the tubular basal membrane (TBM). The α3(IV), α4(IV), and α5(IV) chains are detectable in GBM and the TBM of distal and collecting tubules [43]. Whether these defects predispose people to tubulointerstitial fibrosis and cyst formation is unknown.

Our data suggest that mutations of *COL4A4-A5* are associated with a wide spectrum of kidney abnormalities, providing a dataset of patients showing different renal phenotypes. Disturbed collagen type 4 chains may affect kidney structure and predispose the patients to developing several kidney structural abnormalities, including glomerular sclerosis, cortical calcification in the absence of abnormal calcium–phosphate homeostasis, and fibrocystic kidneys. Thus, similar genetic landscapes may underlie different renal entities, whose clinical onset is variable and includes proteinuria and declined renal function of unknown cause. The reasons that may drive patients to develop different clinical pictures are currently unknown; as a speculation, the site of mutation of COL4 genes, additional genetic factors, comorbidities, and environmental factors may contribute to causing the type of renal disease.

## Figures and Tables

**Figure 1 genes-14-00764-f001:**
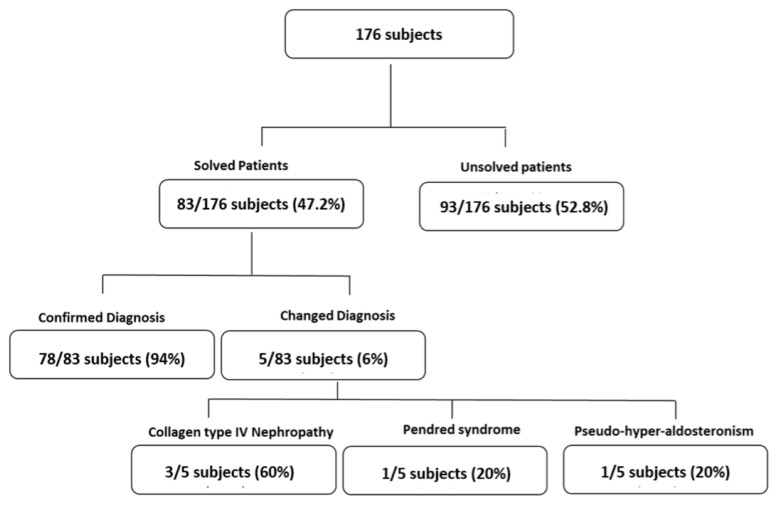
The cohort of patients undergoing NGS analysis, including patients that underwent confirmation and change/perfection of diagnosis after molecular screening.

**Figure 2 genes-14-00764-f002:**
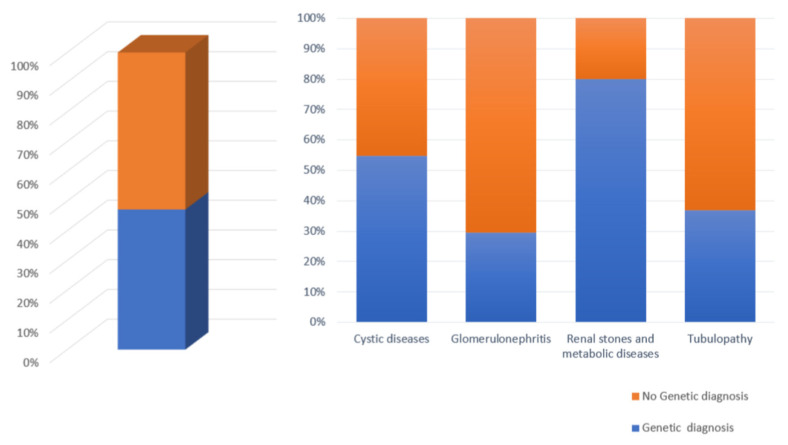
NGS analysis results of the patient cohort, consisting of 176 individuals with inherited kidney disorders. Percentage of genetic diagnoses reached for each category of kidney disease.

**Figure 3 genes-14-00764-f003:**
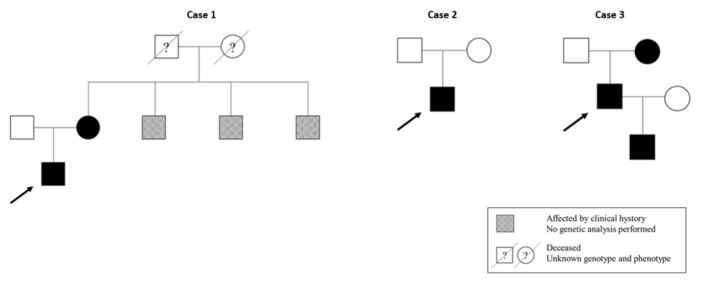
Genealogical trees of *COL4* patients. The arrows indicate the index cases.

**Figure 4 genes-14-00764-f004:**
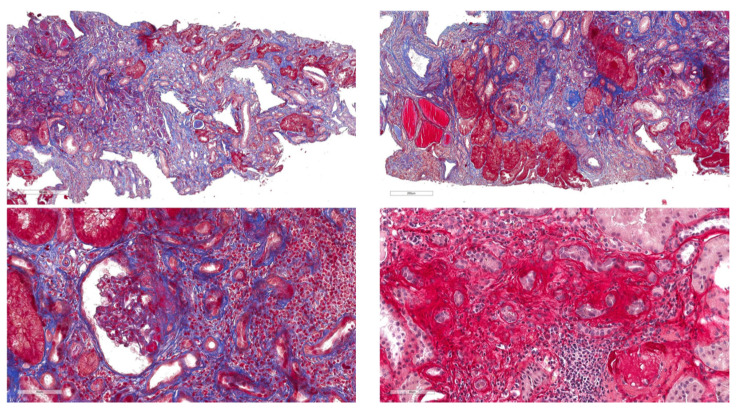
Case 1 kidney biopsy. Light microscopy and immunohistochemistry revealed a classic FSGS with IgM and C3 mesangial deposits, a picture shared with his mother; immunohistochemistry revealed increased deposition of extracellular matrix.

**Figure 5 genes-14-00764-f005:**
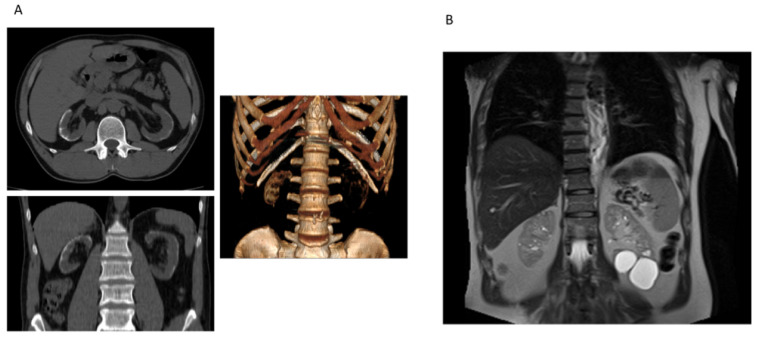
(**A**) Case 2 computed tomography (CT), in axial (upper) and coronal (low) planes, showing cortical and circumferential calcifications. The image on the right shows that the renal cortex has a density similar to the bone, further indicating the diagnosis of cortical nephrocalcinosis. (**B**) Case 3 abdomen CT showing subcentimetric cysts on the right and larger cysts on the left kidney.

**Figure 6 genes-14-00764-f006:**
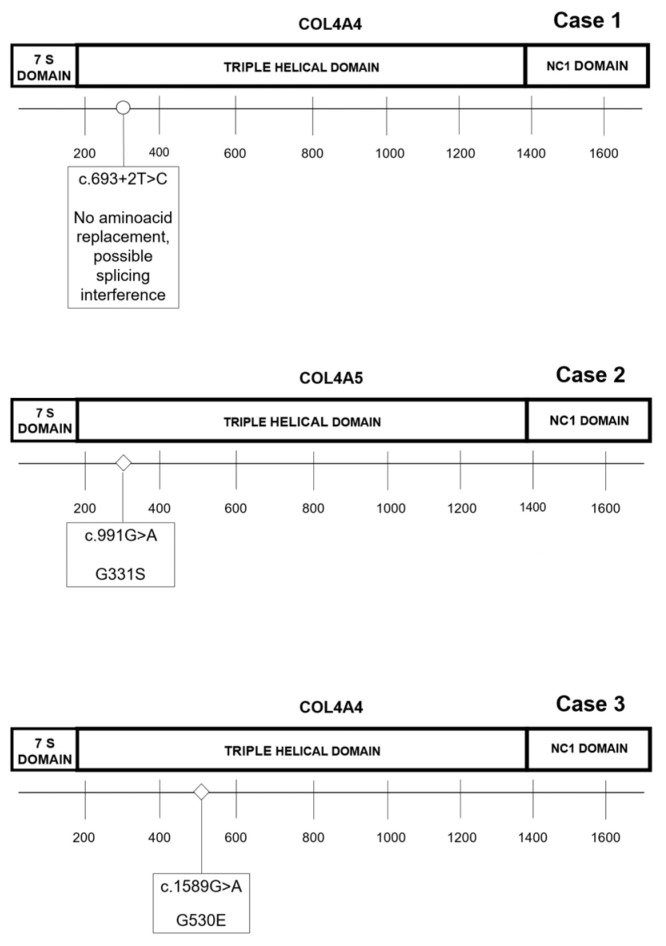
DNA variants of *COL4* patients and resulting protein change.

**Table 1 genes-14-00764-t001:** Main features of patient cohort. The cohort consisted of individuals with different inherited kidney disorders. Age, gender, clinical diagnosis, and diagnostic rate are listed in the table.

Total of Patients Analyzed	176
Patients Analyzed with Nephroplex	107
Patients Analyzed with Clinical Exome Sequencing	69
Males	82
Females	94
Average Age (years)	40.4
Minimum Age (years)	18
Maximum Age (years)	79
Patients with Cystic Disorders	104
Positive Genetic Analysis	57
Patients with Glomerulonephritis	29
Positive Genetic Analysis	8
Patients with Renal Stones and Metabolic Diseases	5
Positive Genetic Analysis	4
Patients with Tubulopathies	38
Positive Genetic Analysis	14

**Table 2 genes-14-00764-t002:** Clinical features of COL4 patients. FSGS, focal segmental glomerulosclerosis.

Proband	Gender	Age (Years)	Hematuria	Proteinuria	eGFR (mL/min/ 1.73 m^2^)	Family History	Renal Histology	Mode of Inheritance
Case 1	male	43	+	+	60	+	FSGS	AD
Case 2	male	49	-	-	13	-	Unavailable	Sporadic case
Case 3	male	59	+	+	39	+	Unavailable	AD

**Table 3 genes-14-00764-t003:** Pathogenicity of variants. For all variants ACMG classification and ClinVar reports are shown.

	GENE	VARIANT	AMINO ACID CHANGE	ACMG	CLINVAR	SIFT	Mu PRO	PANTHER	PhD-SNP	SNP n GO	SPLICE AI
CASE 1	COL4A4	c.693+2T>C	None	Likely pathogenic	Not reported	Not applicable	Not applicable	Not applicable	Not applicable	Not applicable	0.89 delta score as donor loss
CASE 2	COL4A5	c.991G>A	p.Gly331Ser	Pathogenic	Not reported	Affecting protein function (score 0.00)	Decrease Stability	Probably Damaging	Disease	Disease	/
CASE 3	COL4A4	c.1589G>A	p.Gly530Glu	Likely pathogenic	Not reported	Affecting protein function (score 0.00)	Decrease Stability	Probably Damaging	Disease	Disease	/

## Data Availability

Row data can be obtained after request to the corresponding author.

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
