# Peer review of "Next-Generation Sequencing (NGS) Analysis Illustrates the Phenotypic Variability of Collagen Type IV Nephropathies"

_genes, 2023, doi:10.3390/genes14030764_

Round 1

Reviewer 1 Report

In this report, Zacchia and colleagues report the results of NGS-based genetic analysis of a cohort of 176 individuals with suspected inherited kidney disease. They find variants judged to be disease-causal in 85/176 (48.3%) of individuals, encompassing the major categories of genetic forms of kidney disease. They highlight five individuals in whom the genetic findings revealed a clinically unexpected diagnosis of type IV collagen (COL4A) nephropathy, and argue that their study illustrates the value of NGS-based genetic diagnostics for patients with kidney disease and the heterogeneous phenotypic spectrum of COL4A nephropathy. 

The manuscript is overall clearly organized and grammatically written, although there are several typos (e.g., Section 3.1, ‘collage type IV nephropathy’). The clinical value of genetic testing for patients with kidney disease is a relevant topic for the Genes readership. However, as the authors themselves note in the manuscript, there have already been many publications on the varied phenotypic spectrum of individuals with COL4A variants. Moreover, the authors do not provide strong evidence for the pathogenicity of multiple variants they deem diagnostic. Thus, both the originality of the manuscript and the robustness of its data are uncertain at this time. Including additional analyses comparing the findings in the cohort of patients tested here, as well providing stronger evidence for the pathogenicity of the COL4A variants reported, would help address these limitations and make the manuscript more suitable for publication. 

Reviewer 2 Report

The manuscript of Zacchia et al. is an original article presenting the results of the whole exome sequencing on 176 individuals with inherited kidney disorders. The highlight is on the collagen type IV genes. These cases are further characterized. The introduction provides a sufficient background for a person that is not acquainted with the topic. The methods and inclusion criteria are in general well described, but some issues need clarification (see below). The cases are clinically well characterized. The figures are of good quality and they help to understand the content. Discussion and conclusion relate to the result.

Major comments:

1. In the abstract and introduction the diagnosis change is mentioned. For me, it was not clear that the clinical diagnosis was different that the molecular one until the results (I thought that there was another molecular diagnosis with different methods). Please clarify it.

2. Please briefly characterize your cohort in terms of age, sex, etc. , as such data are available

3. Was it an exome-based panel that has been performed? I assume it was not a Sanger-based panel

4. It would be more precise to write e.g. WES than NGS

3. English editing would improve the clarity

Minor comments:

1. Please replace "mutations" with "(pathogenic/likely pathogenic) variants"

2. Figure 6-please put all variants on one illustration

Round 2

Reviewer 1 Report

I thank Zacchia and colleagues for their efforts to revise their manuscript. However, although the revised version has additional methodological detail, it still has many typos (e.g., ‘patients cohot’ in section 3.1, ‘ochlea and eye’ in the Introduction). Moreover, while the authors state that they classified the COL4A variants identified using the ACMG/AMP 2015 guidelines, they do not provide supporting criteria for their classifications. In addition, as the authors themselves note in the manuscript, there have already been many publications on the varied phenotypic spectrum of individuals with COL4A variants, such that the originality of the study remains unclear. These issues must be addressed in order for the manuscript to be more suitable for publication.

Round 3

Reviewer 1 Report

I appreciate Zacchia and colleagues' efforts to revise their paper per my comments. However, the novelty of their findings remains low, and there remains relatively scant evidence supporting that the variants identified qualify as pathogenic or likely pathogenic under the ACMG/AMP guidelines (rather than variants of uncertain significance). These issues need to be further addressed in order for the manuscript to merit publication. 

Round 4

Reviewer 1 Report

I appreciate Zacchia and colleagues revising their manuscript per my comments. Their revisions have satisfactorily addressed my comments and assuming the editors and any other relevant parties agree, I think this article is suitable for publication.